# Qualitative Insights into Vaccine Uptake of Nursing Staff in Long-Term Care Facilities in Finland

**DOI:** 10.3390/vaccines11030530

**Published:** 2023-02-23

**Authors:** Anna-Leena Lohiniva, Idil Hussein, Jaana-Marija Lehtinen, Jonas Sivelä, Suvi Hyökki, Hanna Nohynek, Pekka Nuorti, Outi Lyytikäinen

**Affiliations:** 1Finnish Institute for Health and Welfare, Mannerheimintie 166, 00270 Helsinki, Finland; 2Health Sciences Unit, Faculty of Social Sciences, Tampere University, Kalevankatu 4, 33520 Tampere, Finland

**Keywords:** COVID-19 vaccine, long-term care facilities, Theoretical Domains Framework, behavior change

## Abstract

Vaccine hesitancy and refusal have undermined COVID-19 vaccination efforts of nursing staff. This study aimed to identify behavioral factors associated with COVID-19 vaccine uptake among unvaccinated nursing staff in long-term care facilities (LTCF) in Finland. Methodology: The study was based on the Theoretical Domains Framework. Data were collected through qualitative in-depth interviews among nursing staff and managers of LTCFs. The analysis was based on thematic analysis. We identified seven behavioral domains, with several themes, that reduced the staff’s intention to get vaccinated: knowledge (information overload, inability to identify trustworthy information sources, lack of vaccine-specific and understandable scientific information), beliefs about consequences (incorrect perceptions about the vaccine effectiveness, and lack of trust in the safety of the vaccine), social influences (influence of family and friends), reinforcement (limited abilities of the management to encourage vaccination), beliefs about capabilities (pregnancy or desire to get pregnant), psychological factors (coping with changing opinion), and emotions (confusion, suspicion, disappointment, and fatigue). We also identified three behavioral domains that encouraged vaccine uptake: social influences (trust in health authorities), environmental context and resources (vaccination logistics), and work and professional role (professional pride). The study findings can help authorities to develop tailored vaccine promotion strategies for healthcare workers in LTCFs.

## 1. Introduction

COVID-19 vaccination is critical in reducing severe outcomes of the disease [1,2,3], which highlights the need to ensure sustained and targeted vaccination programs among healthcare workers (HCWs) in long-term care facilities (LTCFs). Vaccination can protect both HCWs and LTCF residents who are particularly vulnerable to severe outcomes [4,5,6,7]. As of June 2021, a notable proportion of COVID-19-associated deaths (40%) occurred among LTCF residents in Finland [7], similar to many European countries [8].

Vaccine hesitancy remains a global phenomenon, particularly when new vaccines are introduced [9], and has been a major obstacle to COVID-19 vaccine uptake globally [10,11]. Vaccine hesitancy is a complex and context-specific phenomenon, varying across time, place, and vaccine type [12,13]. It can be influenced by environmental factors such as physical availability, affordability, willingness to pay, geographical accessibility, ability to understand (language and health literacy), and the ability of immunization services to provide vaccines. It can also be influenced by cultural, social, and behavioral factors, including trust in the effectiveness and safety of vaccines, trust in the system that delivers them, in the reliability and competence of healthcare services and professionals, and the motivations of policymakers who decide which vaccines are included in the vaccination program [12,13,14]. Vaccine hesitancy is much more common than vaccine refusal (objection to vaccines) [15].

COVID-19 vaccine hesitancy and vaccine refusal among HCWs have been hot topics in Finland and elsewhere throughout the pandemic. Initially, the Finnish COVID-19 vaccination strategy identified HCWs as one of the priority groups for vaccination, and accordingly, they were offered vaccines from the beginning of 2021. Previous surveys showed that the voluntary uptake of the COVID-19 vaccine among HCWs was similar to the vaccine coverage in the general population (72.5% for second dose). However, coverage was lowest (less than 70%) among assistant nurses, most of whom were working in LTCFs [16].

Several studies explored the reasons for COVID-19 vaccine hesitancy among HCWs in LTCFs. A study conducted in the US identified personal beliefs and a lack of trust in authorities as being linked with COVID-19 vaccine hesitancy among nursing staff of LTCFs [17,18,19], whereas another study identified fear of side effects as the main reason for hesitancy among HCWs [20]. Most studies exploring hesitancy in LTCFs were quantitative surveys, but some qualitative studies were also conducted. A UK study identified attitudes towards the COVID-19 vaccine as diverse, and found that elements of hesitancy persisted even after receipt of the vaccine [21], whereas another qualitative study in Turkey pointed out that vaccine hesitancy among HCWs was linked with the fear and lack of confidence in the vaccines and inconvenience in accessing the vaccines [22].

The use of frameworks and theories of behavioral insights (BI) in understanding COVID-19 vaccine hesitancy among HCWs and creating solutions to promote vaccination has been uncommon, although BI can provide a structured and systemic approach to understanding people’s behavior. BI can uncover behavioral drivers that may act as barriers or facilitators to behaviors. BI can also ensure that interventions that aim to modify behaviors are evidence-driven and designed to address behavioral triggers [23,24,25]. This paper describes a qualitative study that used a behavioral framework called the Theoretical Domains Framework (TDF) to identify factors that influence hesitancy and refusal of COVID-19 vaccination among nursing staff in LTCFs in Finland.

## 2. Materials and Methods

This qualitative study was initiated during outbreaks caused by the SARS-CoV-2 Alpha and Beta variants in LTCFs in the Helsinki metropolitan area in October 2021 [26]. During the time of the outbreak investigations, all LTCF residents had received the first dose of the COVID-19 vaccine, and most had received the second dose. The investigators discovered that although the nursing staff of the LTCFs under investigation had been offered COVID-19 vaccines, several staff members remained unvaccinated.

The study utilized the Theoretical Domains Framework (TDF) as its conceptual framework, which assigns factors influencing behavior into 14 different domains including knowledge, skills, social influences, social and professional role and identity, reinforcement, beliefs about capabilities, beliefs about consequences, behavioral regulation, environmental context and resources, psychological factors, emotions, goals, intentions, and optimism [27,28,29,30]. A question guide with open-ended and semi-structured questions covering the TDF domains was developed to learn about the reasons nursing staff chose not to take the COVID-19 vaccine (Appendix A). Another question guide was developed for managers to elicit their understanding of the reasons nursing staff chose not to take the vaccine (Appendix A). The question guide for nursing staff was also translated into English and Somali.

### 2.1. Participant Recruitment

The study participants included nursing staff and managers of LTCFs. The inclusion criteria for the nursing staff included being a registered nurse or an assistant nurse working in an LTCF and having experienced at least one COVID-19 outbreak among the facility residents. Managers were recruited to learn about the hesitancy and the refusal of the nursing staff, as the number of nursing staff who volunteered for the study was considered too low to reach data saturation. The inclusion criteria for managers included having managed an LTCF during a COVID-19 outbreak and having personally discussed with nursing staff their reasons for not taking the COVID-19 vaccine.

Nursing staff were first recruited to participate in the study during COVID-19 outbreaks in October and November 2021. Recruitment continued in November and December 2021 through various social media sites. Managers were recruited to participate in the study from March to May 2022 through regional infectious disease doctors who were asked to the share the invitation with members of their professional mailing lists in their region.

All 16 unvaccinated nursing staff who agreed to participate were interviewed. A total of 60 managers volunteered for the interview but 23 could not be reached at the time of recruitment and 11 managers changed their minds about participating in the interview. From the remaining 26 managers, the team selected participants based on maximum variation [31] including managers from both public and private LTCFs, and managers with various years of working experience. The sample size determination was based on data saturation; no new ideas or perceptions appear in the discussions [32].

### 2.2. Data Collection

Two persons experienced in qualitative interviewing carried out the in-depth interviews (IH, ALL) in October and November 2021. Almost all interviews were conducted in a private room of the LTCF. Two interviews were conducted over the phone. The date and the time were selected based on the availability of the participants during their working day. Most were interviewed during their break time. The interviews lasted approximately 60 min, but there were also a few short interviews (about 20 min) and one long interview (almost two hours). All interviews with the managers were conducted by phone (by ALL) from March to May 2022 and lasted approximately one hour each. All interviews were audio recorded with the permission of the study participants. Two nursing staff members refused the recording. Their interviews were documented via short notetaking during the interview and were expanded into long interview notes on the same day. All interviews started with a verbal consent followed by a set of open-ended and semi-structured questions listed in a question guide. All interviews were conducted in Finnish with the exception of one interview conducted in Somali and two in English.

### 2.3. Analysis

The interviews were transcribed verbatim by professional translators. Interviews in English and Somali were translated into Finnish by the interviewers (ALL, IH). The researcher (ALL) carried out a thematic analysis beginning with a familiarization process in which the researcher read the transcripts several times to obtain an overall understanding of the data. This was followed by an inductive coding process within each domain separately to allow new ideas and concepts to emerge from the data [33]. Based on the initial coding, the researcher developed a codebook that was shared with another team member (IH). The process continued by developing subcategories and themes for each domain [34]. Once the domains were analyzed, the themes were shared with the other team members. Any discrepancies in coding and themes were discussed until a consensus was reached. In the final stage, the team members jointly reviewed the themes across the domains and their connections to establish a final interpretation to explain the domains (IH, JML, JS, HN, OL). Only saturated themes were included in the final interpretation of the data. The researchers used NVIVO software to assist with the coding process.

## 3. Results

The results section describes the characteristics of the study participants followed by determinants that discourage and facilitate the uptake of the COVID-19 vaccine. The determinants can be found in Figure 1.

### 3.1. Characteristics of Study Participants

The nursing staff who participated in the study included 16 unvaccinated individuals working in LTCFs in Helsinki’s metropolitan area. Over half (10/16, 62%) were in the age group of 26–50 years. All of them were female. The length of their working experience ranged from 1 to 13 years (median, 8). The ethnic backgrounds included Finnish, Estonian, Russian, Somali, Thai, and Nepali. Half (8/16) expressed having decided not to take the vaccine (refusers), whereas the other half were undecided (hesitant).

The managers who participated in the study included 20 LTCF managers who had experienced COVID-19 outbreaks among residents in their facility. They were located in 8/20 healthcare districts. The majority (14/20, 70%) came from the Helsinki metropolitan area. The managers were highly experienced, 65% (13/20) had >20 years of working experience; range, 1–42 years (median, 15 years).

### 3.2. Information Sources of COVID-19 Vaccines

Participants were asked where nursing staff members sought COVID-19 vaccine-related information. All participants (nursing staff and managers) stated that the main information sources were traditional media, social media, and social networks including friends and family.

Most nursing staff participants (regardless of whether they were vaccine-hesitant or vaccine refusers) highlighted that COVID-19-related information would not have aided in their decision to take or refuse the vaccine. Most of them explained that the information received at the workplace was related to vaccine logistics. Some participants clarified that they had been told the type of vaccine available but no further information.


*“They just told us that the vaccine was now available, and we could take it during working hours at the city vaccination sites.”*

*(Assistant nurse, private sector)*


### 3.3. Factors Discouraging Uptake of COVID-19 Vaccine

Seven domains acted as barriers to the uptake of the COVID-19 vaccine among nursing staff in LTCFs: knowledge, beliefs about consequences, social influences, reinforcement, psychological factors, beliefs about capabilities, and emotions.

#### 3.3.1. Domain: Knowledge

##### Information Overload

Many nursing staff participants (hesitant and refusers) reiterated having received an overabundance of information as vaccine-related information was shared simultaneously from various sources, often with contradictory messages and changing information, which led to feelings of not knowing enough about the vaccine.


*“I am getting tired. Too much information, too confusing. I don’t feel I am getting a clear understanding of the vaccine.”*

*(Assistant nurse, public sector)*


##### Inability to Identify Reliable Information Sources

Several nursing staff participants (hesitant and refusers) also highlighted being confused about whom to trust when experts were providing a wide range of views. For example, one respondent explained that different public health experts gave entirely different opinions about the aim of the vaccination strategy at the same time on national television. Another respondent pointed out that a university professor contradicted the views of the daily COVID-19 briefing of the government.


*“Different doctors said different things about corona vaccines. In the end, you can’t trust anyone. You decide to follow your instincts. I felt the corona vaccine could be a risk and I am not willing to take that risk.”*

*(Assistant nurse, private sector)*


##### Lack of Vaccine-Specific Information

Some nursing staff participants (hesitant and refusers) explained that despite excessive information, they lacked credible scientific information about the vaccine such as detailed information about the ingredients and vaccine-specific side effects. A few participants also expressed lacking an understanding of how mRNA vaccines function and the difference between mRNA vaccines and other vaccines. One respondent who had refused the vaccine highlighted that she did not know how to choose the most appropriate vaccine for herself.


*“I went to the vaccination point in the city and asked them to show me the package so that I know what the vaccine contains and what the potential side effects are. Any medicine has this description. They told me that this information was not available. This is when I decided not to take the vaccine.”*

*(Assistant nurse, private sector)*


##### Lack of Understandable Information

Most managers explained that the staff had difficulties with COVID-19 vaccine-related information as the information that they received from the regions, municipalities, the Finnish Institute for Health and Welfare (THL), and other sources was often lengthy with complex language and concepts.


*“We sometimes have to read the information a number of times. I am not surprised that the nursing staff does not always grasp what is being communicated. When the Astra Zeneca vaccine was halted, I found many staff members not understanding the situation. They wondered why and how that vaccine differed from the other available vaccines. There was general worry around the vaccine and no available information to reassure.”*

*(Manager, private sector)*


Most managers agreed that their employer had not disseminated much vaccine-related information Some managers thought there was enough information available outside of the workplace.


*“I think there is so much information, so we do not need to take the role of disseminating vaccine-related information.”*

*(Manager, public sector)*


#### 3.3.2. Domain: Beliefs about Consequences

##### Perceptions Undermining the Efficacy of the Vaccine

All vaccine-hesitant nursing staff participants explained that one reason for not taking the vaccine was because they did not view it as beneficial. They had heard from others that it did not prevent the infection and transmission of the virus, which they thought was the main function of the vaccine.


*“We have learned that even if you are vaccinated you can transmit the virus. What kind of vaccine is this? How does it make sense to take the vaccine?”*

*(Assistant nurse, private sector)*


All nursing staff participants who had refused the vaccine mentioned lack of vaccine efficacy as one reason for their position. Their views were often based on personal observation. One explained having witnessed vaccinated staff getting infected with the coronavirus and non-vaccinated staff remaining negative.


*“We thought that the vaccine can prevent virus transmission. But it does not. Many of our residents were vaccinated and they still got the virus. I see no reason to take the vaccine.”*

*(Assistant nurse, private sector)*


Several managers confirmed staff members claims that even if they took the vaccine, they could bring the virus from outside.


*“Many of my nursing staff believe strongly that the vaccine was a failure and with or without the vaccine the situation remains the same. I only had some staff members who didn’t take the vaccine but this is where the discussions were focused.”*

*(Manager, private sector)*


Nursing staff participants (hesitant and refusers) reiterated COVID-19 vaccination as a major disappointment. Despite high vaccination coverage in the LTCFs, all infection prevention and control measures remained in place including the mandatory use of the mask. Managers explained that many staff members were highly upset to realize that taking the vaccine would not allow them to discontinue the use of a mask.


*“Speaking with my colleagues, nothing has happened although all residents and many nursing staff members have received the vaccine. It is a big disappointment that we have to carry on using the masks.”*

*(Assistant nurse, public sector)*


Some nursing staff members (refusers) believed that the vaccine was not needed after being infected with the virus as natural immunity was reached. Others believed that a coronavirus infection resulted in only a limited risk of contracting COVID-19, which did not necessitate taking the vaccine.


*“First, they told me to wait 6 months because I had corona. After that, I did not feel like taking it. What would I benefit of the vaccine? I have immunity.”*

*(Assistant nurse, private sector)*


##### Lack of Trust in the Safety of the COVID-19 Vaccine

All vaccine-hesitant nursing staff discussed their worry about potential side effects of the COVID-19 vaccine, such as severe symptoms, long-term or unknown side effects. Often the side effects were discussed in general terms without specifying them. Most of their fears were linked to their own vaccine experiences or the experiences of others related to the COVID-19 vaccine, earlier flu vaccinations, or even with routine immunization in childhood.


*“Honestly, I have very bad experiences from my childhood. I got so sick when I got my vaccination. This is what my mom used to say. I am thinking that I will have some strong side effects should I take the [COVID-19] vaccine.”*

*(Assistant nurse, private sector)*


Nursing staff participants who refused the COVID-19 vaccine expressed their fear of side effects by referring to specific symptoms such as high fever, physical disabilities, or narcolepsy.


*“I only need to think about the swine flu pandemic and I get worried. I don’t want to get severe side effects and be unable to continue my everyday life. I am thinking of all these people who got narcolepsy from that vaccine.”*

*(Assistant nurse, public sector)*


Most managers highlighted that they had staff members who worried about infertility or other reproductive health complications and were, therefore, reluctant to take the vaccine. Some managers mentioned that their staff members believed that the vaccine could not be safe as it was developed so fast. Most managers also agreed that many fears about side effects stemmed from staff sharing personal negative experiences they had with other vaccines.


*“I had some staff members that believe in these circulating rumors about infertility. They are not based on any real information, but they are real and powerful.”*

*(Manager, private sector)*


#### 3.3.3. Domain: Social Influences

##### Influence of Family and Friends

Nursing staff participants (hesitant and refusers) highlighted that family and friends influenced their COVID-19 vaccine decision in many ways. Some of them explained that continuous discussions with friends who were against the vaccine swayed their opinion about the vaccine over time. Others mentioned that those who were against the vaccine usually had more convincing arguments than those who were for it. One participant (refuser) cited that her husband convinced her not to take the vaccine. In another case (refuser), it was the mother who convinced the participant not to take the vaccine.


*“None of my friends have taken the vaccine. It was clear to me that I would not take it either.”*

*(Assistant nurse, public sector)*


#### 3.3.4. Domain: Reinforcement

##### Limited Management Encouragement

All nursing staff participants who refused the COVID-19 vaccine highlighted having had no discussions with the management about vaccination. Many of them pointed out that this reinforced the idea that the taking the vaccine was unimportant. In addition, many of the participants stated that the COVID-19 vaccine was introduced as voluntary, which made them feel it was unimportant to get vaccinated. Moreover, a few participants mentioned that their manager had not taken the vaccine, which again reinforced their idea that the vaccine was not a necessity.


*“Nobody has asked me about the vaccine. We don’t speak about it.”*

*(Assistant nurse, public sector)*


All vaccine-hesitant nursing staff participants also pointed out that vaccine-related discussions with the management had been limited, but they believed that the situation reflected the choice that they were given to take the vaccine or to refuse to take it. It was not seen as a discouragement.


*“My boss told us that the vaccine was available. He did not say anything more. I felt perhaps he wanted us to take the vaccine, but it was for me to decide that.”*

*(Assistant nurse, public sector)*


##### Refraining from Discussing Vaccine Uptake

All managers perceived discussions about the COVID-19 vaccine as challenging as, according to the Finnish law, the employer is not allowed to pressure their staff members to take the vaccine. Some managers pointed out that they received instructions from higher management levels not to talk about the vaccine at all except to provide information about where and how the staff can take it. For example, one manager explained having avoided the staff room so as not to be engaged in any vaccine-related discussions.


*“We haven’t spoken about it. I don’t know who took the vaccine. It is a personal matter.”*

*(Manager, private sector)*


##### Consciously Maintaining a Neutral Position

Some managers made a conscious decision to remain neutral about the vaccine by transparently advocating for staff members to decide for themselves.


*“I made clear to my staff that I am neutral, and I am not planning to give them advice.”*

*(Manager, public sector)*


##### Using Indirect Influence

Other managers intended to influence staff opinion by acting as an example and taking the vaccine.


*“I made sure that all my staff knew that I took the vaccine. It is important to be an example.”*

*(Manager, public sector)*


A few managers clarified that they themselves were against taking the COVID-19 vaccine which made them reluctant the discuss the topic with their staff.


*“I am not convinced about the vaccine myself.”*

*(Manager, private sector)*


#### 3.3.5. Domain: Psychological Factors

##### Coping with Changing Opinions

Some nursing staff participants who refused the COVID-19 vaccine explained that their decision-making process was influenced by their early position that they were against the vaccine. They could not change their opinion out of embarrassment. Others pointed out that they were unable to change their position although they “rationally felt” that they had been wrong in not accepting the vaccine.


*“I had some inaccurate ideas about the pandemic. I did not understand that it would reach such a massive scope. But because I had already made up my mind not to take the vaccine, I was not ready to change my position.”*

*(Assistant nurse, public sector)*


Some managers confirmed these views by explaining that they had staff members who would have been willing to take the vaccine later during the pandemic but felt too embarrassed publicly change their minds.

#### 3.3.6. Domain: Beliefs about Capabilities

##### Physical Capability

Half of the nursing staff participants who refused the vaccine mentioned their main reason for the refusal was pregnancy or a desire to get pregnant later. Some of them explained having followed their doctor’s advice not to take the vaccine. Others said they had not received a clear opinion from their doctor on whether to take the vaccine, which made them decide against the vaccine. Several respondents knew that the vaccine guidelines had changed over time and that pregnant women were also recommended to take the vaccine, but they did not feel comfortable doing so, for they did not trust that it was safe, or they were simply unmotivated to do it.


*“I don’t want to take the risk of not having children. I am still young.”*

*(Assistant nurse, public sector)*


Managers acknowledged that some nursing staff members had not taken the vaccine because of pregnancy or because they feared reproductive health problems such as infertility.


*“Those who were pregnant refused the vaccine even after it was recommended. They wanted to play it safe.”*

*(Manager, private sector)*


#### 3.3.7. Domain: Emotions

##### Confusion, Suspicion, Disappointment, and Fatigue

Nursing staff participants (hesitant and refusers) were aligned regarding the emotions that the COVID-19 vaccine evoked. They explained that, in the beginning, emotions were linked with confusion and suspicion about the vaccine, as information was rapidly changing and often contradictory. Some of them also cited various conspiracy theories that caused concern early in the vaccine rollout, such as vaccines being a government plan to eliminate certain ethnic groups. Later in the pandemic, when sufficient vaccine coverage had been reached, the vaccine evoked disappointment and overall pandemic fatigue as infection prevention and control practices remained the same. Many participants mentioned the continued use of face masks as a particular disappointment.


*“This has been an emotional rollercoaster with different feelings but always strong ones.”*

*(Assistant nurse, private sector)*


### 3.4. Factors That Encourage the Uptake of the COVID-19 Vaccine

Three domains were identified as facilitators of the uptake of the COVID-19 vaccine among nursing staff in LTCFs: social influences, environmental context and resources, and professional role.

#### 3.4.1. Domain: Social Influences

##### Influence of Family and Friends

A few hesitant nursing staff participants highlighted that their family members and friends had an overall positive opinion towards the uptake of the vaccine which, in turn, made them more confident to take the vaccine.


*“My husband thinks I should take the vaccine. I am considering it.”*

*(Assistant nurse, private sector)*


##### Trust in Authorities

Some vaccine-hesitant nursing staff participants discussed trust toward health authorities by emphasizing their knowledge, abilities, and professionalism to control the pandemic. The same participants emphasized understanding that the health authorities did not want to harm them.


*“I know that any vaccine that is offered to us is well researched and planned. I know nobody wants to harm us.”*

*(Assistant nurse, private sector)*


#### 3.4.2. Domain: Environmental Context and Resources

##### Easy Logistics

Most nursing staff participants clarified that vaccine logistics had worked fine regardless of whether the vaccine had been delivered in their facility or at a public vaccination site. Participants described the vaccine system as easy, simple, and clear. Some participants highlighted that on-site vaccination was the easiest way to get the staff vaccinated. Several managers agreed with that view pointing out that on-site vaccination made vaccination easy, particularly for those who were unmotivated to book a vaccination appointment or physically go to another location. A number of nursing staff participants for whom Finnish was not their first language noted that booking a vaccination appointment was not difficult.


*“Vaccines were made available and instructions on how to take them were given. There was nothing unclear about it or difficult.”*

*(Assistant nurse, private sector)*


#### 3.4.3. Domain: Professional Role

##### Professional Pride

Several nursing staff participants (hesitant and refusers) expressed feelings of guilt for not having taken the vaccine. They acknowledged being in a special position to protect populations at risk.


*“I know that as frontline workers, we must take the vaccine, but I am afraid. It makes me feel so guilty.”*

*(Assistant nurse, public sector)*


Some nursing staff participants who refused the vaccine cited understanding their medical role in protecting vulnerable people but suggested that LTCFs can intensify infection prevention and control measures rather than rely on the vaccine to protect the residents.


*“I make sure to practice hand hygiene procedures and I am always wearing a mask. This is the way I protect the residents. The vaccine does not play a major role.”*

*(Assistant nurse, private sector)*


Several managers believed that the majority of their nursing staff had a strong professional identity as a caretaker of vulnerable people that motivated vaccine uptake.


*“The great majority of my staff believes that nursing staff has a special need to take the vaccine.”*

*(Manager, public sector)*


## 4. Discussion

The study identified seven behavioral domains (TDF) that may be associated with nursing staff hesitation and refusal to take the COVID-19 vaccine, including knowledge, beliefs about consequences, social influences, reinforcement, beliefs about capabilities, psychological factors, and emotions. The study also identified three TDFs that were associated with a more positive attitude towards COVID-19 vaccination including social influences, environmental context and resources, and professional role. These findings can be used to develop interventions to encourage COVID-19 vaccination among nursing staff of LTCFs, including future booster vaccinations and new vaccinations for future epidemics and pandemics.

Our study showed that both nursing staff and managers had a lack of knowledge about COVID-19 vaccines. The reasons for this included pandemic-related information overload, lack of vaccine-specific, clear information, as well as the inability to identify reliable information sources, which all appeared to be barriers to vaccine uptake. Information overload and, more broadly, “the infodemic”, meaning an overabundance of information—some accurate and some not—that occurs during an epidemic, has been a global challenge [35,36,37,38]. Many studies highlighted that the infodemic contributed to stress, confusion, and fatigue among HCWs [39,40,41]. The WHO framework to manage the infodemic highlights the importance of providing information that is science-based, can reach the intended target audience, and is clear enough to allow people to make an informed choice [35]. This highlights the need to develop an information and knowledge-sharing system that reaches the nursing staff of LTCF and provides information that is understandable, so that the nursing staff regard themselves as informed enough to decide for themselves whether or not to take the vaccine. Around the world, there have been various interventions to help manage information, such as a one-pager “quick sheet” that includes continuously updated guidelines, policies, and practical information to serve as a reference tool for HCWs [42], or the creation of an instrument to measure information overload, which consequently informs strategies to manage information levels in a clinical setting [43].

Addressing the infodemic provides an opportunity to tackle concerns about the efficacy and safety of the vaccine, which we identified as major barriers to vaccine uptake. Both safety and efficacy of the vaccines are common concerns related to any type of vaccine worldwide among the general population and among HCWs [44,45,46,47]. A recent study in the UK showed that COVID-19 vaccination intentions can be strengthened through a simple messaging intervention that utilizes perceived vaccine response efficacy [48]. When developing such safety messages, it is important to think about how to frame them. Framing refers to the process by which people develop a particular conceptualization of an issue based on words, images, phrases, and a presentation style [49]. Frames influence the same information and the processing of that information including the intention to act and they change over time differently, highlighting the need to modify safety and efficacy messages as the epidemic evolves [49,50].

Our study also showed that the influence of social networks was significant. Friends and family members who shared their negative opinions about the COVID-19 vaccine discouraged vaccination and, likewise, positive opinions encouraged it. Previous studies indicated that being exposed to positive attitudes, frequently discussing vaccinations with family and friends/peers, or wanting to comply with their behavior increased the likelihood of HCWs getting vaccinated [47,51]. A study from China showed that having a social network with whom to communicate about the COVID-19 vaccine alone made HCWs less hesitant towards the vaccine [52]. Another study among nursing home staff in the US highlighted that the staff preferred to see local community members and people similar to themselves being vaccinated to improve their confidence in the safety and effectiveness of the vaccine. As such, giving priority allocation of the vaccine to LTCF staff before community members may discourage vaccination [53]. This also shows the importance of partnering with community members when promoting vaccines among nursing staff. Behavioral insights interventions that target nursing staff, residents, and their family members jointly should be considered. Moreover, it is important to consider social influences on the nursing staff, as, in our study, the nursing staff reported using the same information sources as the general population. Accordingly, public health officials should also consider messages targeted to HCWs in LTCFs when promoting vaccines to the general public [54,55,56,57]. Public health authorities and policymakers may also consider including the social networks of the nursing staff in their strategies to deliver behavioral insights interventions to encourage vaccine uptake.

Lack of management reinforcement was a barrier to the uptake of the COVID-19 vaccine. Managers agreed with that view and acknowledged having little ability to encourage vaccination, as they were instructed by the authorities to offer vaccination to volunteers and not to pressure their staff, which makes their position delicate even if they wish to promote the vaccine. Previous vaccination behavior studies have found reinforcement to be consistently associated with higher acceptance of the COVID-19 vaccine [58,59,60]. Policy-level and organizational-level interventions are required to find an appropriate role for the management in vaccine promotion efforts that allows them to reflect a positive attitude towards the vaccines, but not create pressure on the staff. In addition, there is a need to build the capacity of the management to allow them to use evidence-based interventions that have been proven to encourage vaccine uptake, such as various communication techniques including personalized communication, affective messaging, risk framing, norm framing, persuasive messaging, expert claims, and information messaging [61].

Our study also identified facilitators that may be used in interventions to further increase vaccination acceptance and to motivate HCWs to take the vaccine, such as professional pride, which reflects an understanding of the professional responsibility to protect others. A study in Italy similarly discovered that perceived professional responsibility was associated with higher vaccination acceptance which could potentially be leveraged at the healthcare organization level [60]. Many studies indicated that HCWs who provide direct care for patients have higher vaccine acceptance than other HCWs, which may also reflect professional responsibility [61,62,63,64]. Accordingly, professional responsibility and professional pride are concepts that could be further utilized when promoting COVID-19 vaccines among the nursing staff in LTCFs.

We also identified trust in authorities as a facilitator for accepting the COVID-19 vaccine, but only among participants who were hesitant. This is consistent with previous studies that found that mistrust in government and public health bodies was associated with lower vaccine acceptance [55,56,57]. Trust in the processes and systems that develop, approve, and monitor the safety of vaccines, as well as the people and agencies who recommend and endorse vaccination, cannot be overlooked when designing vaccine promotion interventions. Accordingly, continuous trust-building interventions during crises are of utmost importance [65,66,67]. Community engagement strategies that are based on the long-term collaborative process have been identified as more effective in building trust than short-term interventions such as public campaigns [68]. National and international public health agencies must prioritize facilitating a broad understanding of these processes among the public through frequent, consistent, and visible communication, including engaging with representatives from communities and populations who are disproportionally affected by the pandemic to leverage their knowledge, skills, and expertise and to listen to their concerns [69]. Trust can be also increased by working with the media and social media influencers aiming at decreasing sensationalism and instead portraying a more honest picture in their reporting, which are lessons learned from previous epidemics [58]. Moreover, building trust through public figures has been identified as an effective tactic [70].

We had limitations in our study. Recruitment of non-vaccinated nursing staff during the outbreak was extremely difficult as many of them were reluctant to participate in our interviews. To increase the number of study participants, we used additional recruitment channels such as nursing schools and social media channels. We included managers of the LTCF as target audiences to increase the overall sample size and for triangulation purposes. The sample size was not large enough to provide separate recommendations for the vaccine-hesitant and vaccine refusers though they are known to be distinct groups in many ways [71]. Instead, the study outlined factors that collectively influenced nursing staff who had not taken the vaccine and linked them to recommendations that can be jointly used to promote vaccines among nursing staff. The study findings may also have been influenced by social desirability bias [72], as the interviews were conducted during an outbreak that was a particularly stressful situation for nursing staff participants. We did not address the type of vaccines specifically. At the time of the study, mRNA vaccines were offered in Finland. and they caused discussion about their safety across the globe [73], which may have influenced our results. Lastly, we did not ask about the flu vaccine history of the respondents, which could have described better the attitude of the respondents towards vaccines in general.

The study provided theory-based evidence on how to develop behaviorally informed interventions to improve vaccine acceptance among nursing staff and managers in LTCFs and other homecare types of facilities. Such strategies typically rely on self-acceptance and internalized understanding of the phenomenon and are likely to lead to sustained behavior change [74].

Accordingly, they can also help to reduce the need to rely on vaccine mandates such as semi-mandatory vaccination for HCW that was implemented in Finland from January 2022 onwards or the semi-mandatory influenza vaccine policy that came into effect in 2017 [75,76]. Semi-mandatory means that vaccines are obligatory but can be avoided through certain procedures such as testing or the use of protective barriers. Both policies have raised negative public discussion in the country.

This study identified behavioral barriers and facilitators for COVID-19 vaccination acceptance among nursing staff in LTCFs in Finland that may be used to develop interventions to promote vaccination. The study also helps decision-makers navigate this complex issue by identifying behavioral determinants that can be leveraged to develop vaccine promotion strategies.

## Figures and Tables

**Figure 1 vaccines-11-00530-f001:**
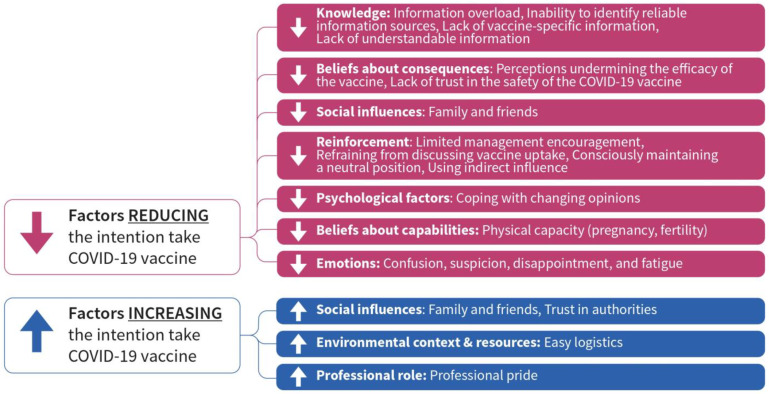
TDF domains that increase and decrease uptake of COVID-19 vaccine among nursing staff in LTCF.

## Data Availability

The data presented in this study are available on request from the corresponding author. The data are not publicly available due to privacy.

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
