# Peer review of "Qualitative Insights into Vaccine Uptake of Nursing Staff in Long-Term Care Facilities in Finland"

_vaccines, 2023, doi:10.3390/vaccines11030530_

Round 1
Reviewer 1 Report
Does the introduction provide sufficient background and include all relevant references?
The Introduction section of the manuscript provides a comprehensive theoretical background of the research. This section provides a detailed view of the very current study context and summarizes relevant literature related to the topic.
Is the research design appropriate?
This research design is appropriate and accurately described.
Are the methods adequately described?
The Materials and Methods section is described in detail. The three different subsections make the manuscript more comprehensible to the reader: Annexes S1 and S2 help follow the next search steps.
Are the results clearly presented?
In the Results section, the results of the analysis are described exhaustively. I suggest a final summary table listing the seven identified behavioural domains and a few keywords that characterize each domain. This way, such a section would be more accessible and fast reading.
Are the conclusions supported by the results?
The manuscript provides essential conclusions and implications in the “Discussions” section.
I suggest merging the Discussions and Conclusions sections because a paragraph of just five lines would be unstructured. Alternatively, expand the Conclusions section by discussing possible futile developments in research.
Author Response
We thank the reviewer for the comments. See our responses to each comment below.
Does the introduction provide sufficient background and include all relevant references?
The Introduction section of the manuscript provides a comprehensive theoretical background of the research. This section provides a detailed view of the very current study context and summarizes relevant literature related to the topic.
Is the research design appropriate?
This research design is appropriate and accurately described.
Are the methods adequately described?
The Materials and Methods section is described in detail. The three different subsections make the manuscript more comprehensible to the reader: Annexes S1 and S2 help follow the next search steps.
Are the results clearly presented?
In the Results section, the results of the analysis are described exhaustively. I suggest a final summary table listing the seven identified behavioural domains and a few keywords that characterize each domain. This way, such a section would be more accessible and fast reading.
Response: We already included a summary table in our initial submission that includes TDF domains and linked keywords. But we have now added more details to it. See Figure 1.
Are the conclusions supported by the results?
The manuscript provides essential conclusions and implications in the “Discussions” section.
I suggest merging the Discussions and Conclusions sections because a paragraph of just five lines would be unstructured. Alternatively, expand the Conclusions section by discussing possible futile developments in research.
Response: Thank you for the suggestion to merge the conclusion section with the discussion. Accordingly, we have made the changes in the manuscript.

Reviewer 2 Report
The manuscript “Qualitative insights into vaccine uptake of nursing staff in 2 long-term care facilities in Finland” aims to “identify behavioural factors associated with COVID-19 vaccine uptake 11 among unvaccinated nursing staff in long-term care facilities (LTCF) in Finland”. The methodology, a qualitative study that utilized the Theoretical Domains Framework (TDF) provided valuable insights that can help health authorities to improve vaccination in this group of health workers. The small number of interviews, the sample size, is an important limitation of the study.
Author Response
We thank the reviewer for the comments. See below our responses to each comment:
The manuscript “Qualitative insights into vaccine uptake of nursing staff in 2 long-term care facilities in Finland” aims to “identify behavioural factors associated with COVID-19 vaccine uptake 11 among unvaccinated nursing staff in long-term care facilities (LTCF) in Finland”. The methodology, a qualitative study that utilized the Theoretical Domains Framework (TDF) provided valuable insights that can help health authorities to improve vaccination in this group of health workers. The small number of interviews, the sample size, is an important limitation of the study.
Response: We fully agree with the reviewer. Small sample size is a common problem in qualitative research studies. We list the small sample size as a limitation in our discussion. P 11-12 lines 541-548
Recruitment of non-vaccinated nursing staff during the outbreak was extremely difficult as many of them were reluctant to participate in our interviews. To increase the number of study participants we used additional recruitment channels such as nursing schools and social media channels. We included managers of the LTCF as target audiences to increase the overall sample size and for triangulation purposes. The sample size was not large enough to provide separate recommendations for the vaccine-hesitant and vaccine refusers though they are known to be distinct groups in many ways [71].
Reviewer 3 Report
Anna-Leena et al., seek to understand factors that prompt COVID vaccine hesitancy and refusal among nursing staff in long-term care facilities (LTCF) in Finland through interviewing nurses (unvaccinated) and managers at these facilities. This manuscript reports common themes that emerge from these interviews and suggest steps that could be taken to increase confidence and uptake of vaccines.
Understanding the drivers of vaccine hesitancy/refusal is important from a public health standpoint. The data provides useful insights, though I am of this opinion that this study could have delved deeper. Comments are as below-
1. Nursing staff that agreed and received the vaccine should have also been interviewed. It is important to understand what factors promote vaccine acceptance.
2. It should be mentioned what COVID vaccine(s) were being offered in Finland at the time of the study.
To this point, it is important to address whether the hesitancy was reflective of the use of mRNA vaccines (new platform) as opposed to older-traditional platforms?
3. Was the vaccine hesitancy/refusal specific to COVID vaccines or extended to others as well? Prior history of vaccination should have been inquired. What about flu vaccine?
4. The paper could provide quantitative analysis, such as percentage of interviewed individuals that said “x” was a factor in their decision. The aim would be to give the readers a glimpse into which factors are more important than others.
Author Response
We thank the reviewers for their comments. Our responses to each comment can be found below:
Anna-Leena et al., seek to understand factors that prompt COVID vaccine hesitancy and refusal among nursing staff in long-term care facilities (LTCF) in Finland by interviewing nurses (unvaccinated) and managers at these facilities. This manuscript reports common themes that emerge from these interviews and suggest steps that could be taken to increase confidence and uptake of vaccines.
Understanding the drivers of vaccine hesitancy/refusal is important from a public health standpoint. The data provides useful insights, though I am of the opinion that this study could have delved deeper. Comments are as below-
- Nursing staff that agreed and received the vaccine should have also been interviewed. It is important to understand what factors promote vaccine acceptance.
Response: As our study emerged during the time of outbreaks in LTCFs from the observation that many nursing staff members remained unvaccinated, the focus of the study was on those who refused or were hesitant. We agree that it is important to also study nursing staff who agreed to receive the vaccine, but it would have been difficult to cover that in this study. We were unable to include both groups in our study because of resource and time constraints. It is clear that the topic requires a separate study.
- It should be mentioned what COVID vaccine(s) were being offered in Finland at the time of the study. To this point, it is important to address whether the hesitancy was reflective of the use of mRNA vaccines (new platform) as opposed to older-traditional platforms.
Response: We did not ask about the type of vaccine in our interviews although at the time of our study both mRNA and adenovirus vector vaccines were available in Finland (Obach et al ). We have added that as a limitation.
- Was the vaccine hesitancy/refusal specific to COVID vaccines or extended to others as well? Prior history of vaccination should have been inquired. What about flu vaccine?
Response: The study was focused on hesitancy/refusal of COVID-19 vaccines. However, although influenza vaccination is required for healthcare staff, we do not have data about the participants’ influenza vaccine history. We have added this as a limitation.
The paper could provide quantitative analysis, such as the percentage of interviewed individuals that said “x” was a factor in their decision. The aim would be to give the readers a glimpse into which factors are more important than others.
Response: We followed a typical qualitative data analysis and reporting that is constructed around themes. As this is not based on a random sample, quantity in % would not reveal the importance of the themes. With the qualitative analysis, we can only state that the emerging themes are present.